# Positive Energy Districts: Identifying Challenges and Interdependencies

Savis Gohari Krangsås [1,*], Koen Steemers [2], Thaleia Konstantinou [3], Silvia Soutullo [4], Mingming Liu [5], Emanuela Giancola [4], Bahri Prebreza [6], Touraj Ashrafian [7], Lina Murauskaitė [8] and Nienke Maas [9]

1 Department of Architecture and Planning, Norwegian University of Science and Technology (NTNU), 7491 Trondheim, Norway
2 Department of Architecture, University of Cambridge, Cambridge CB2 1PX, UK; kas11@cam.ac.uk
3 Faculty of Architecture, Delft University of Technology (TU Delft), 2628 BL Delft, The Netherlands; t.konstantinou@tudelft.nl
4 Department of Energy, Centro de Investigaciones Energéticas, Medioambientales y Tecnológicas (CIEMAT), 28040 Madrid, Spain; silvia.soutullo@ciemat.es (S.S.); emanuela.giancola@ciemat.es (E.G.)
5 School of Electronic Engineering, Dublin City University, 9 Dublin, Ireland; mingming.liu@dcu.ie
6 Faculty of Electrical and Computer Engineering, University of Pristina, 10000 Pristina, Kosovo; bahri.prebreza@uni-pr.edu
7 Faculty of Architecture and Design, Özyeğin University, Istanbul 34794, Turkey; touraj.ashrafian@ozyegin.edu.tr
8 Laboratory of Energy Systems Research, Lithuanian Energy Institute, LT-44403 Kaunas, Lithuania; Lina.Murauskaite@lei.lt
9 Department of Strategy and Policy, The Netherlands Organization for Applied Scientific Research (TNO), 2595 AD The Hague, The Netherlands; nienke.maas@tno.nl
* Correspondence: savis.gohari@ntnu.no

**Abstract:** Positive Energy Districts (PED) are areas within cities that generate more renewable energy than they consume, contributing to cities' energy system transformation toward carbon neutrality. Since PED is a novel concept, the implementation is very challenging. Within the European Cooperation in Science and Technology (COST) Action, which offers an open space for collaboration among scientists across Europe (and beyond), this paper asks what the needs for supporting the implementation of PEDs are. To answer this, it draws on Delphi process (expert reviews) as the main method alongside the literature review and also uses surveys as supplementary methods to identify the main challenges for developing PEDs. Initial findings reveal seven interacting topics that later were ranked as highest to the lowest as the following: governance, incentive, social, process, market, technology and context. These are interrelated and interdependent, implying that none can be considered in isolation of the others and cannot be left out in order to ensure the successful development of PEDs. The resources that are needed to address these challenges are a common need for systematic understanding of the processes behind them, as well as cross-disciplinary models and protocols to manage the complexity of developing PEDs. The results can be the basis for devising the conceptual framework on the development of new PED guides and tools.

**Keywords:** Positive Energy District; challenges; COST Action; governance; needs; tools; market; participation; collaboration

## 1. Introduction

Europe aims to be a global role model in energy transition and reducing its carbon footprint, thereby moving towards sustainable development. Since cities are the main centers of greenhouse gas production, using 65–70% of global energy and producing 70–75% of global emissions, European cities are urged to control and reduce emissions from their buildings and districts [1] In this regard, the European Union introduced the program "Positive Energy Districts (PEDs) for Sustainable Urban Development" to initiate and

support the planning, deployment and replication of 100 positive energy neighborhoods by 2025 [2]. These districts are a key part of creating a comprehensive approach to sustainable urbanization and dealing with technological, spatial, regulatory, financial, legal, social and economic perspectives [3,4].

Due to the novelty and multifaceted nature of this urban concept [5], there are multiple interpretations and definitions that make it challenging to implement, evaluate, compare, or replicate these districts. Saheb et al. [6] analyzed four zero energy community projects to identify a global framework that these communities could implement. Brozovsky et al. [7] reviewed the latest articles published on climate-friendly neighborhoods to contextualize this concept and identify the main targets, needs and gaps. Hearn et al. [8] proposed a framework that integrated energy justice and quality of life in implementing a PED to guarantee global well-being to all the residents. These districts aim to develop livable and innovative spaces that facilitate the energy transition towards decarbonization and meet climate, social and economic objectives.

In the framework of the SET Plan Action 3.2, JPI Urban Europe and the EERA Joint Program on Smart Cities, PEDs are defined as:

"Energy-efficient and energy-flexible urban areas or groups of connected buildings which produce net zero greenhouse gas emissions and actively manage an annual local or regional surplus production of renewable energy. They require integration of different systems and infrastructures and interaction between buildings, the users and the regional energy, mobility and ICT systems while securing the energy supply and a good life for all in line with social, economic and environmental sustainability."

Participating PED program partners have agreed that this PED reference framework offers a common baseline across all countries while ensuring flexibility regarding local conditions for PEDs at the same time. In this regard, the concept of PEDs is evolving and still needs to be refined, advanced and redeveloped to be demonstrated, implemented and replicated [9].

Implementation of the PEDs requires a deep understanding and consideration of cities' contextual conditions, policies, priorities, strategies, resources and solutions. Knowledge, skills and technologies are needed for planning, designing, implementation and monitoring, as well as replication and mainstreaming of PEDs. Even though many European cities are leading transitions to low-carbon energy, there is no joint definition, roadmap and guidelines to ensure the actual feasibility of PED designs, mainly because cities are in planning or early implementation stages [2]. In this regard, there is still a need for identification of main requirements for implementing PEDs and understanding the interconnection and synergies between these requirements. The COST Action "CA19126—Positive Energy Districts European Network" (PED-EU-NET) (https://pedeu.net/, accessed on 16 September 2021) started in November 2020) contributes to this as well as the deployment of PEDs in Europe by facilitating knowledge, experience exchange and collaboration in research and innovations among cities, industry and research organizations.

This CA acknowledges that the main challenge to propel Europe towards its goal is to open up the innovation process to all active players so that knowledge can flow freely across the entire economic and social environment [10]. The deployment of PEDs requires innovations in multiple domains encompassing interconnected technological, social, cultural, political, spatial, financial and regulatory aspects. Each domain has its own set of embedded challenges that need to be tackled in order to foster the innovation process.

There are both technical and non-technical challenges to creating an overarching vision and framework for PEDs. On the one hand, the aim is to define generalizable tools, guidelines and targets. On the other hand, it is necessary to respond to local stakeholders, approaches and conditions. Based on Europe-wide consultation with city representatives, urban stakeholders and national experts, the PED reference framework has categorized the challenges in terms of technological, spatial, regulatory, financial, legal, ecological, social and economic perspectives [11]. Although no ranking was implied, it was understandable

that the technical challenges were mentioned first given the energy and emissions targets outlined previously. The survey of case study PEDs provides a ranking of the success factors and challenges according to those involved in implementing the projects [2]. Based on this information, the issues that were considered the most important, across both the success factors and the challenges, can be ranked as follows (Figure 1):

(1) Governance (politics, policy, regulations and city administration);
(2) Social (stakeholder and citizen engagement);
(3) Market (funding, markets and business models); and
(4) Technical (energy and urban integration).

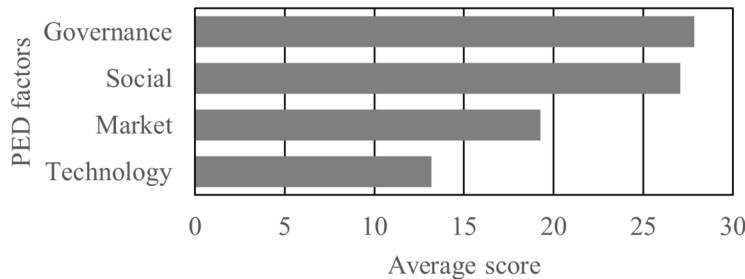

**Figure 1.** Votes for perceived success factors and challenges of PEDs, equally weighted and combined into the top four principal categories, to reveal their ranking of importance (after [2]).

As shown in Figure 1, the governance and social factors ranked highest and were very close to each other, with a noticeable gap before the market and technology factors. The purpose of this paper is to define these and other challenges systematically, with more descriptive detail and to identify interdependencies between them.

One of the objectives of the CA is to generate insights into the development of new guides and tools for optimizing the design, operation, financing and opportunities of PEDs. This paper is aligned with this objective, and aims to first define exiting challenges systematically and with more descriptive detail in order to identify interdependencies between them, and second to discover what are needed (e.g., in terms of tools, guides, resources, methods, guidelines, etc.) for overcoming the identified challenges.

## 2. Methods

### 2.1. A Three-Stage Approach

The openness of COST Action (https://www.cost.eu, accessed on 16 September 2021) to all fields of science and technology as well as all different sectors, institutions and countries, has served as a great foundation for this paper to collect different experts' perspectives on the needs for implementing PED. Thus, the methods used in this paper build on the COST Action's PED European Network (PED-EU-NET), representing key professions, a range of cultures and climates, disciplines and experiences from different sectors, but mainly academia's perspectives (160 members from 38 countries: Albania, Austria, Belgium, Bosnia and Herzegovina, Bulgaria, Croatia, Cyprus, Czech Republic, Denmark, Estonia, Finland, France, Germany, Greece, Hungary, Ireland, Israel, Italy, Latvia, Lithuania, Luxembourg, Malta, Moldova, Montenegro, Netherlands, North Macedonia, Norway, Poland, Portugal, Romania, Serbia, Slovenia, Spain, Sweden, Switzerland, Turkey, and United Kingdom).

The first stage of the approach was to carry out a literature review of the existing state-of-the-art research related to PEDs. This is summed up in the Introduction and Results Section of this paper. The second stage was to adopt the Delphi method, which is a structured communication technique that relies on a core panel of experts [12,13]. This method was our main source of data, which were used to brainstorm and identify the key challenges to implementing PEDs and what responses are needed to overcome them. Debates with the expert panel (from Norway, United Kingdom, the Netherlands, Spain,

Ireland, Kosovo, Turkey and Lithuania) and continuous detailed individual discussions led to a consolidation of the definition of these challenges and the interdependencies between them. A third stage consisted of a survey of the wider PED-EU-NET membership and a deeper literature review, as a supplementary method, to examine and validate the results of the Delphi method, determine the ranking of the key challenges and whether additional factors should be included. In addition, the survey and internal COST Action meetings gave us an opportunity to collect the perspectives and feedbacks of the other countries/disciplines on the existing challenges, increasing the representativeness of more regions of Europe such as Hungry, Portugal, Germany and Switzerland.

While initiatives for creating PEDs are launched in many European cities, due to the novelty of PEDs, the knowledge in this field is still limited, which has been a methodological challenge for our research. In addition, the complex and multifaceted nature of PEDs make the question of how to implement PEDs unanswered.

### 2.2. The Delphi Method

The 10 authors of this paper—experts drawn from the research and development (R&D) community of PED-EU-NET—formed the core panel. The aim of the panel was to identify the key challenges that confront the implementation of PEDs. The Delphi method provided a structured communication and decision-making technique by which the panel of experts can address the questions, led by a facilitator (in this instance, this was Dr. Gohari Krangsås). Furthermore, the Delphi method is well suited as a research instrument when there is incomplete knowledge about a problem or phenomenon [12,13]. Given the early stage of development of PEDs, and the very few case studies identified as being in operation and realized to date—only two in Europe [14]—this method was particularly appropriate and timely.

The principle of the method is that findings from a structured group of experts will be more accurate than those from an unstructured, random, unrepresentative or undefined group. By working with R&D experts with a breadth of 'disinterested' knowledge of PEDs, supported by literature reviews, we tried to avoid the potentially distorting effects or the gaps in knowledge of different special interest groups, such as political, business or citizen representatives. We were then able to compare and contrast our findings with those of others available in the literature, and where there may be vested interests, or where a more unstructured approach has been adopted.

In this research, the Delphi method involved six months of iterations with the expert panel, where initial decisions were recorded, revisited and reviewed before finally being agreed upon. The first of these communications consisted of a brainstorming session using 'Padlet' (https://padlet.com/, accessed on 16 September 2021), a collaborative web platform hosted by the facilitator and via which panel members could share and organize content to a virtual bulletin board. In total, four communications were held between November 2020 and June 2021. The first meetings identified seven key topics/challenges and the final one resulted in a detailed definition of each.

### 2.3. Survey

Once the Delphi method had revealed the seven challenges for implementing PED, a survey was prepared using 'Mentimeter' (https://www.mentimeter.com/, accessed on 16 September 2021, an online interactive polling tool. This survey consisted of three sections: (1) to rank the seven factors in order of importance from a drop-down menu, (2) to score each of the challenges in terms of the strength of agreement and (3) to use free text to identify other factors to note. This survey was sent to the members of the PED-EU-NET project during May 2021 with a response rate of 15%. Even though the number of responses was fewer than we expected, it supported our qualitative research objective. Our main focus in this research was to gather in-depth insights on the topic, which is not well understood yet, in order to formulate a deep understanding about the main needs and challenges of implementing PEDs, rather than establishing generalizable facts about this

topic through math and statistical analysis. However, in future, a quantitative study can be conducted to draw a more general conclusion from this research.

## 3. Seven Topics/Challenges Identified by the Delphi Method

From the iterative Delphi method, the panel experts identified the following seven challenging topics, in provisional order of importance, representing what we need to overcome to support the implementation of PEDs:

1.  Governance: a need for new and innovative forms of collaborative governance;
2.  Incentives: a need for right (social and environmental) drivers and motivators;
3.  Social: a need for local community's support and engagement;
4.  Process: a need for integrated planning and decision-making approaches;
5.  Market: a need for an appropriate market design and business model;
6.  Technology: a need for balancing energy demand and supply systems;
7.  Context: a need for considering regional and local differences.

The four categories of Good and Ceseña [15]—governance, social, market and technology perspectives (see Figure 1)—also emerged in our findings and in the same rank order. This result provided a degree of confirmation and reassurance that the two methods were compatible. However, our finding highlights the importance of three other considerations, including incentives, process and context, which rank amongst the other four categories (Figure 2).

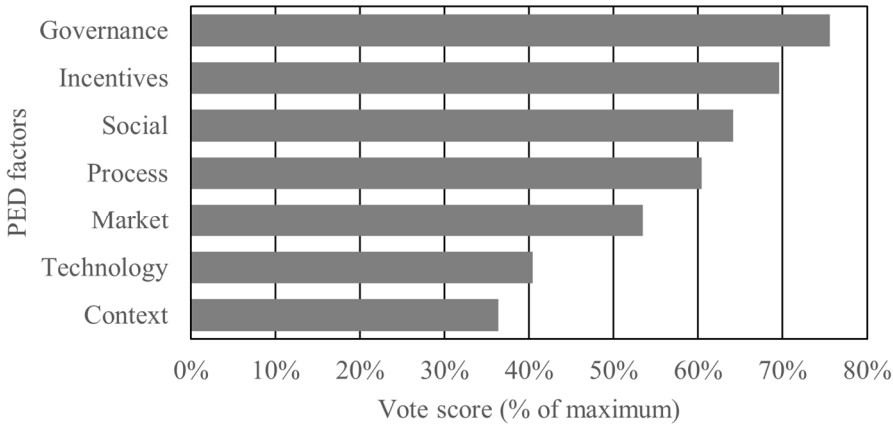

**Figure 2.** Survey results of ranking PED challenges showing the percentage of scores based on voting scores of 7 for the first ranking to 1 for the last ranking out of the maximum score.

It also became evident in the Delphi events that there were multiple interactions between these seven topics. The relationships between them will be discussed in Section 5. Each PED challenge is described briefly below before exploring the potential interdependencies.

### 3.1. A Need for a New and Innovative Forms of Collaborative Governance

While technological innovation is a necessary to make PEDs, the challenge is not primarily on technology, but on service transformation, management, policy and improvement [16]. The bulk of research effort and journal space is devoted to documenting the unintended consequences, paradoxes and shortcomings of the traditional governing mode. It is acknowledged that managing all the aspects of a smart city often lies beyond the capabilities and mandates of the traditional government. Accordingly, new and innovative forms of governance are needed, in which various stakeholders, including citizens, take part in the planning and decision-making process, share control over development initiatives, collaboratively address problems and set priorities to build commitment and ownership of the final planning outcome.

Governance is not the same as governing or government. While 'governing' refers to those social activities, which make a "purposeful effort to guide, steer, control, or manage

(sectors or facets of) societies", 'governance' describes "the patterns that emerge from the governing activities of social, political and administrative actors" [17] (p. 2); and while government centers on formal authority, governance refers to activities backed by shared goals that may or may not derive from legal or formally prescribed responsibilities [11] (p. 4). Thus, governance is associated with network structures, interdependency, trust relations, negotiations and power relations among different actors, which contrasts with the traditional hierarchical government. In this regard, the relational configuration of PED governance can widely vary depending on the actors involved, their role, their impact and the degree of (in)formality of relationships among them. Further, it is subject to change during the networks' lifecycle, which can be described in terms of the three main phases of initiation, emergence and wider implementation or uptake. However, understanding of these features and dynamics as well as the mechanisms of sharing different resources to foster knowledge flows is still limited, and the question of 'which governance model is the best (if any)' is under a lively debate in research and empirical practice [18]. In addition, still there is no practical framework, method or model that can provide us with a better understanding of the overarching functionality of the entire system within which different stakeholders are collaborating. There is no guidance for conceptualizing the governance processes, including the individuals' roles and influences on outcomes, the way their power is exercised and the degree to which the public and their interests should be involved. On the other hand, without good governance, the identified challenges cannot be managed.

To understand what governance systems should be employed to support implementation of PEDs, the existing governance underpinning complex planning systems should be analyzed and evaluated. Potts et al. [19] (p. 13) suggest considering how the system is structured and organized, but also the way in which the structures in the system function. Since different structures and functions of PEDs are interconnected and interdependent, the first step is to deeply understand each identified topic/challenge and the synergy between them in the context of an ever-changing, complex and unpredictable PED system.

### 3.2. A Need for Right Incentives

The adoption of the right incentives is a key issue to achieve PEDs. There are multiple benefits stemming from ordinances creating incentives for renewable energy, including economic, ecological and health benefits. In this regard, local governments have a variety of options for creating incentives to support or subsidize the installation of renewable energy equipment, including offering rebates on purchasing equipment, tax incentives, expedited permitting and others [20]. Incentives can also be used by the private sector to encourage cities, housing associations, households and companies to implement PEDs. The outcome of incentives can be focused on supporting increased deployment of targeted technologies and practices, environmental gains, livability and inclusiveness in districts. As Figure 3 represents, incentives can create additional local jobs in energy business, improving air quality, benefitting public health and preventing energy poverty [20,21].

Providing the right incentives depends not only on the specific energy policies in a specific context, but also on how related markets work, and especially how prices are set in these markets [22]. Policies may seek to alter behavior by offering new and beneficial technology, changing financial and other material incentives, changing attitudes and beliefs with education and information, appealing to basic values or modifying institutional structures that may range from international agreements down to community-level norms and neighborhood organizations [23].

On the other hand, the same solutions are not working in different contexts due to the impacts produced in each city [24]. Incentives are thus contextual and specific to energy efficiency technology, as sizes, costs and/or performance measures, development level of the district, local markets and consumer preferences. In this regard, incentives should be designed tailormade. Accordingly, a robust technical and economic analysis of potential technologies can support successful incentive design [20].

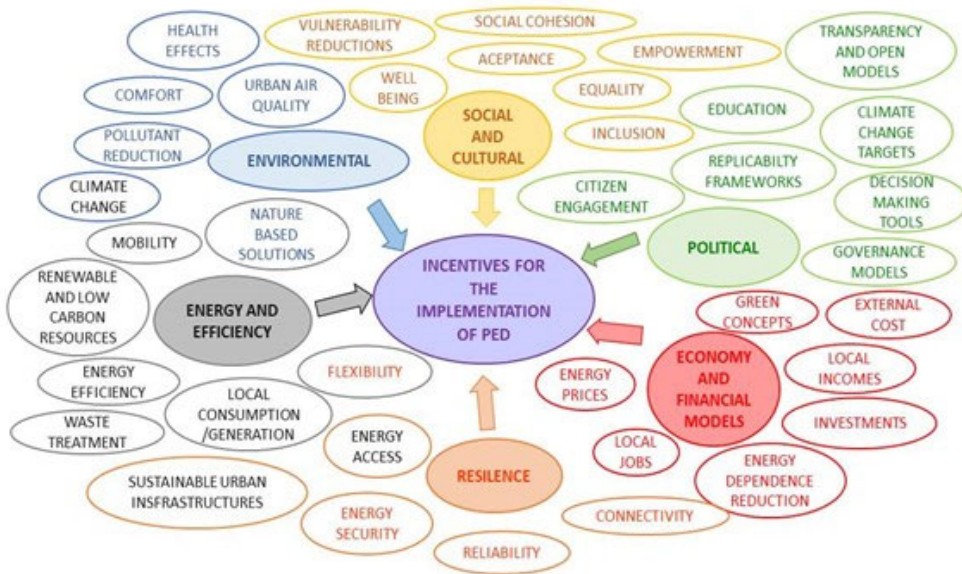

**Figure 3.** Main incentives achieved with the implementation of PEDs.

Högberg and Lind [25] stress the complexity of providing the right incentives, explaining how district heating tariffs may reduce incentives as they make it difficult for a customer to know how profitable a certain energy efficiency investment really is. There have been cases in Sweden where a district heating company, owned by the municipality and using a cost-based tariff, increased prices when energy consumption was reduced. Such actions reduce the incentive for housing companies to invest in energy efficiency [19]. Therefore, it is important to investigate the incentive effects of various pricing principles on the profitability of energy-saving investments.

Cox [20] suggests that providing the right incentives should aim at reducing limitations of long-term and low-interest investment funding schemes, regulatory barriers, absence of economic support/subsidies and unstable policy frameworks. Furthermore, to quantify the success of adequate incentives, it is necessary to consider performance indicators, i.e., the effectiveness of the proposed urban solutions through monitoring and control systems [14] and impact indicators [26]. Finally, Cox [20] also argues that for providing the right incentives all urban stakeholders involved in different phases must be considered, including such as target markets and communities, private sector, utilities and local government institutions.

### 3.3. A Need for the Local Community's Support

There is a widespread consensus on the importance of the citizens' and end-users' role in sustainable energy transitions and changes from passive consumption to active prosumption and engagement [2,27]. Integrating the citizens' perspectives into decision-making and design of services and infrastructure will increase their impact. When citizens are involved in the planning process, their acceptance will be increased, and thus, implementation will be easier. The perspectives of citizens as non-experts about local environment, context and place can (re)discover more sustainable and creative solutions that fit in a specific local context, which might never have been broached or might have been forgotten [28]. As Halachmi and Holzer [29] argue, citizen participation is an important element not only to achieve the democratic governance process, but also to increase government productivity, citizen satisfaction, citizens' trust in government and transparency to make decisions about service levels, procedures and priorities.

In PEDs, consumers should be empowered to drive the transition by optimizing their energy behaviors in day-to-day life, adopting lower carbon transport and heating options, participating in demand response and energy trading activities, investing in energy-efficient technologies and, more importantly, to be engaged in local energy initiatives. In order

to solve the challenges of the present energy systems, the focus should not only be on individual behavioral change, but on system-wide transformation through collective action, which is a successful motor of social transformation [23]. Thus, local citizens can be engaged in community energy systems' supply-side activities, such as collective purchasing of solar panels or collective ownership of wind farms, and also demand-side activities, such as energy conservation, retrofitting of dwellings or energy awareness-raising activities [30].

Engaging local communities in the energy market provides them with a better choice of energy supply and possibility of producing and selling their own energy (as prosumers) with real-time responses to price signals [31]. Koirala and Koliou [32] draw attention to the factors that determine willingness of local citizens to participate in the local energy systems and categorize them as follows: demographic and socio-economic (such as age, education, family situation, home ownership, tax deduction and income), socio-institutional (such as sense of community and trust) and environmental factors (such as ownership of distributed energy resources, resiliency, environmental concern and desire to reduce $CO_2$ emissions). However, Paone and Bacher [33] argue that these factors can result in the uncertainty in the prediction of occupant-related energy behavior, thus creating a gap between actual and predicted energy performance of buildings within a block or district.

Massey and Verma [34], address the following challenges that exist for involving local communities. Firstly, the lack of citizens' and local organizations' engagement in the energy transition and the communication gap between government and community pose a major challenge. Secondly, the absence of appropriate infrastructure for the transparency of strategies/policies and daily engagement of community is a major barrier. Thirdly, shortage of public trust in new energy technologies and a fear of not being able to adapt are challenges. Fourthly, insufficiency of knowledge of citizens about technical topics and policy, and finally, the lack of a model or structure for community engagement make implementation difficult.

These challenges necessitate development of a clear roadmap, which can show the transition to sustainable energy from the perspective of community participation, including policy and regulation, organizational and financial issues, as well as infrastructural development.

### 3.4. A Need for Different Planning and Decision-Making Approaches

The decision-making process involves basically the problem, objectives, alternatives, evaluation and implementation of the decision [35]. By an appropriate planning of these meta-decisions, the decision makers would have more control over the process and can reach better quality decisions with less time and resources invested, thus optimizing the decision-making process [36].

The decision-making process of PEDs is strongly influenced by the complex interconnections between technical, economic and political factors. Even though many scientific publications are dealing with decision-making frameworks and energy planning [37,38], the focus is often on individual decision-making step, and there are no established democratized, multicriteria approaches [39,40].

Some theories assume that policies are set at a certain moment (rational approach) and others assume that concurrent streams of problems, solutions and politics set a policy (garbage can approach). However, in reality, policies result from a series of decisions taken by various actors during a period of time [41]. Decisions are no longer arranged based on a priori order and hierarchy, but on different iterative decision-making rounds. Each round of decision making can change the direction of the match or the rules of the game because new actors can appear, introducing new problems and solutions and a solution for one actor could easily be a problem for another. As a result, decision making is about dynamic combinations of sets of problems and solutions represented by different actors [42] and different exogenous factors (e.g., the political and governmental priorities, contextual differences, etc.).

In the planning of PEDs, where the interests and goals of the government, residents, energy utilities companies, property developers and many other actors are confronted, it is usual, for example, to establish a joint supply of electricity, heat and cooling for achieving efficiency. In this regard, a joint, transparent and structured decision-making process is therefore required to support the entire process from the preliminary design to the operation of the supply concept. Specifically, the district's energy supply should be climate friendly, supply residents with energy at market prices and have the highest possible rate of own consumption [43].

In addition, as highlighted in Section 3.1 in the first challenge "governance", it is widely recognized that social aspects play an essential role in the successful implementation of the PED [44]. In this sense, for example, one should consider that most studies and practical experiences about PED are based on projects in newly built districts, where the planning and integration of innovative solutions are less complex, and the ambition is usually higher [2]. However, buildings in historic districts present particularly challenging characteristics to ambitious energy refurbishment and therefore are usually not considered in PED projects [10] due to the environmental and well-being problems of this kind of districts, coupled with severe regulatory limitations to implement energy efficiency measures and to integrate renewable energy systems (RES) [45].

As concluded by Lyhne [46], decision-making processes are formed by a continuous interaction between policymaking and planning, which is taking place in windows of opportunities rather than formal approvals of plans and policies. Thus, it is not always appropriate to treat policymaking and planning separately, but they should be considered in the interaction with each other; in this interaction, public consultation, systematic environmental analyses and transparency on alternatives are primarily related to choices of planning character.

### 3.5. A Need for Appropriate Market Design

Setting up a PED is a complex process. It involves many stakeholders, each with their interests, constraints and agendas. In the building market, three categories of stakeholders can be identified [47]; policy, community and market. Each category has distinct roles in the development of the PEDs.

Several studies on the barriers to energy efficiency [48] have shown that the lack of an appropriate market can create barriers to implement PEDs. A market cannot function properly due to the imperfect information or incomplete markets (that result in some parties free riding) [15]. Information on how the market is designed and economic aspects would support the municipalities to initiate the development of PED. On the other hand, availability of PED technologies and their implementation depend on the planners and the technology suppliers. Thus, PED's market includes identifying the customer segments for the interventions and the value that the PED will deliver for them. Additionally, it defines the financial process as the activity system creates economic value [48].

Analysis of existing PEDs shows that both funding and feasible business models rank very high among key aspects of PED development [2]. This can be expected because managing different PED actors requires a high degree of coordination [49].

As Wüstenhagen and Boehnke [50] explain, adopting an appropriately designed business model is an important opportunity to overcome some of the key barriers to the market diffusion of sustainable energy technologies. Organizations may be able to convert their supply chains and customer interfaces toward a sustainability focus, but they may not consider links to other business model elements, such as value propositions and financial models in exploring business model transitions [51]. A single company can never meet the needs of a city, nor can a city implement innovative solutions without cooperation with business partners from different sectors [52]. Business models are a tool to define different stakeholders' roles and coordinate their activities and interactions.

In essence, the business models can be seen as an outline that prescribes how the enterprise delivers value to customers, entices customers to pay for value and converts

those payments into profit [53]. By describing the value, a company can offer to one or several segments of customers, and the architecture of the firm and its network of partners for creating, marketing and delivering this value and relationship capital, profitable and sustainable revenue streams are generated [54].

One of the main challenges for PED implementation is that the different sectors' business cases are not integral and tend to work in silos. There is no predefined single business model for the successful development of a PED. Instead, a combination of different business models has to be found for each stakeholder involved [49]. For example, the energy market linked to a smart community implies a distributed and decentralized system operation in which the energy can be generated, stored and distributed by a wide variety of technologies close to the consumption points. The integration of multiple energy vectors represents a great challenge for the management and control of the energy market. The flexibility, reliability and manageability of the market require a high level of effectiveness and automation in the generation and operation of the energy.

Furthermore, structures and policy instruments that mobilize the financing for the investments on PED are required for the implementation. This can be achieved by direct financing interventions to close the gap, as well as long-term perspective in the investment [55]. The best market design should satisfy the short-run efficiency—making the best use of existing resources—and long-run efficiency—promoting efficient investment in new resources [56]. PEDs have the potential for economic sustainability, due to cost efficiency and self-consumption [57].

### 3.6. A Need for Balancing Demand and Supply

The increasing amount of installed distributed renewable generation is transforming the generation side into a more variable and intermittent source of energy. The high randomness produced by these renewable sources can lead to greater fluctuations on the supply-side. In addition, the demand-side is becoming more active, emphasizing the empowerment and engagement of consumers. New concepts are emerging such as prosumers or customers that can produce and supply electricity and thermal energy [58]. "Integrating both factors together can result in demand and supply balancing issues, requiring an optimized adjustment through advanced management and control techniques to avoid losses in the flexibility efficiency and resilience of the energy systems."

Due to the need to deal with uncertain and intermittent output and load shifting of renewable energy systems (RES), energy storage is one of the most critical arguments today [59]. Energy storage technologies can provide needed flexibility and resilience while being more influential at the district level [60]. There are challenges in implementing practical energy storage systems that can result in cost inefficiency and reduced energy performance [61].

The energy balance in PEDs needs to devise effective energy strategies, including power, gas and thermal energy networks [62]. The Set Plan Action 3.2 highlights the necessity of integrated innovative solutions at the district scale [63] to achieve an annual net zero energy import, net zero $CO_2$ emissions and annual local surplus of renewable production. The overall energy balance of a district must consider different factors both in geographical and temporal terms, which are outlined as follows:

- Available natural resources to determine renewable generation.
- District characteristics (density, morphology and contextual factors) to define the typology and seasonality of the energy consumption.
- Available infrastructures to determine the energy vector interactions and control systems.
- Distributed poly-generation, considering the fluctuations produced by renewable systems and possible interconnections with other energy networks.
- Policies and regulations to manage the operations implemented that ensure the stability, accessibility and flexibility of the system.

There is no common methodology for calculating the energy balance of a district. The most usual methods obtain the energy positivity through different indicators such as non-renewable primary energy ratio [64] or on-site energy ratio [65].

To illustrate some key concepts of how different operators can interplay with each other, a schematic diagram is shown below. Figure 4 demonstrates the balance depends on the available elements on both supply and demand-sides by effectively interacting with the storage units as the energy buffer. To reach this goal, integrative urban infrastructures that take advantage of local and renewable resources, ICT systems, efficient innovative technologies, adequate urban planning and the flexibility of the regulatory framework are necessary [66].

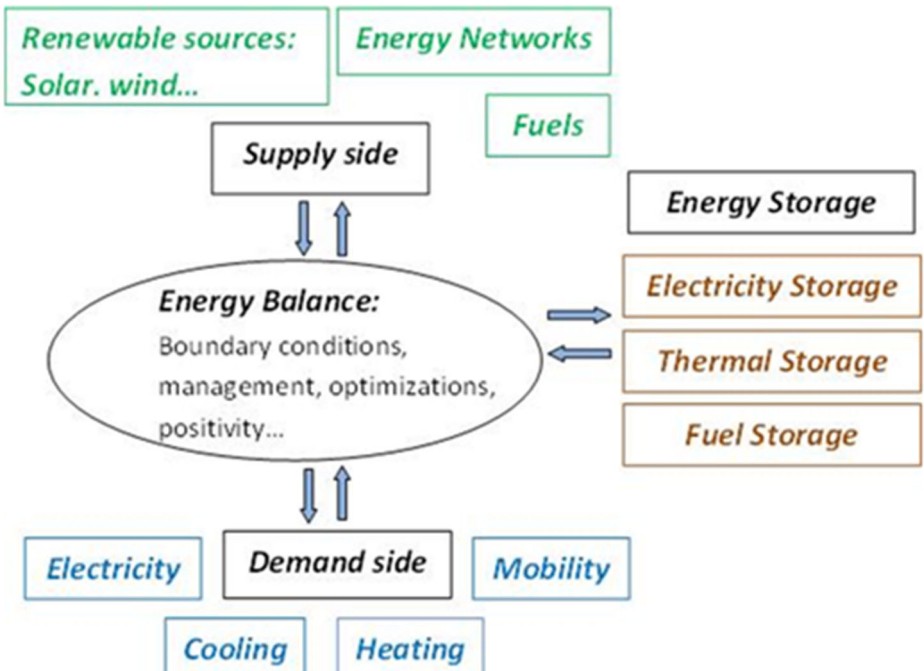

**Figure 4.** Urban flows in the energy balance of PEDs.

To achieve an optimal operation for a multi-vectoral optimization, thermal, electrical and gas flows need to be considered comprehensively together with the influence that these energy vectors exert on other ones [67,68]. To do this, a PED system needs to leverage advanced ICT, big data and automation techniques to obtain high reliability and a low response time on integrated infrastructures, which is technically challenging [69].

It is worth noting that an effective system operation is also influenced by the environment, management system, positivity calculations or regulatory restrictions. Given this, it is essential to set up a common methodology to quantify global energy balance and environmental benefit of a smart urban system through the combination of several performance indicators [14]. In fact, there is no standard at this point to calculate the positivity of an energy balance at the district level, and the energy flows between the buildings and energy systems are very case specific and sometimes complex to understand [70]. Standards can help but may not be user-friendly and interpretable [71]; for instance, which loads/elements should be included in the calculation, which renewable energy technologies should be considered and which primary energy factors should be used are all challenges to be solved.

### 3.7. A Need for Considering Different Contextual Factors

As all the aforementioned challenges have described, the successful implementation of PEDs is highly dependent on a context-specific approach that includes the interactions between regional socio-political, technical, spatial and economic factors.

The Paris Climate Agreement (2015), requiring net zero emissions by the middle of this century, has led to binding interim targets and is supported by numerous building and energy directives [70]. Given that by 2050 the building stock will consist of 85% of old buildings that currently exist, any intervention (whether building a new or refurbishment) must strive to be 'positive energy' in order to meet the overall targets.

The technology and knowledge required to achieved PEDs already exists—put simply, the use of renewable sources for building energy use needs to double and gas usage needs to be halved [55]. However, the emissions reduction ambition remains a challenge in the context of "disappointing levels of improvement in energy efficiency across policy fields" across European regions [72], so it is clear that other contextual factors are at play.

Egger and Gignac [72] identified a key strategy: to develop stronger and diverse narratives that respond to the regional context. These should address the wider value of PEDs, including opportunities for local jobs, industry and competitiveness. A focus on new messages is required—beyond energy or climate imperatives—that is relevant to get the acceptance and participation of local stakeholders. This implies that any generic framework for implementing PEDs requires a nuanced and bespoke interpretation that enables it to respond to the specific locale.

In planning terms, urban contexts provide challenges and opportunities for the development of PEDs. City center neighborhoods are typically dense and historic, with a mix of uses and ownership patterns. This in turn creates different temporal and spatial patterns of energy demand. Whilst one part of a district may require cooling during a summer's afternoon (e.g., a commercial office building) another may need energy for hot water or cooking in the evening (e.g., in housing).

Thus, there are opportunities to balance energy demands across a district and relate them to bespoke renewable energy provision and storage strategies suited to the context. For example, existing technologies, such as ground source heat exchange and combined heat and power (CHP), combined with energy refurbishment may be most appropriate in high-density, geometrically complex or historic urban areas where visual and technical integration is a challenge [73]. Conversely, solar and wind energy strategies are more easily incorporated in new and less-dense neighborhoods where more opportunity is established for solar roofscapes, or space for wind generation can be allocated from the outset of the project [73].

Most studies tend to assume or focus on new districts where the planning and integration of energy systems can be more easily and efficiently achieved compared to established or historic districts [70]. This has some merit in terms of early adoption and pilot projects. More rapid development of PEDs should be pursued in contexts where strategies are especially economical, reducing the overall cost for a region or city. This would help to offset the challenge of creating PEDs in contexts where implementation may be more difficult or costly, and where the lessons learnt from early adopting districts can be subsequently and progressively applied within a timeframe to meet the policy targets.

## 4. The Survey

The survey we conducted for our research not only produced a ranking of which PED factor was considered the most important, as shown in Figure 2, but also asked subjects to score to what extent they agreed about the importance of individual factors (i.e., not in comparison to other factors) (see Figure 5). The results are used to examine how much consistency there is amongst experts for each PED topic/challenge.

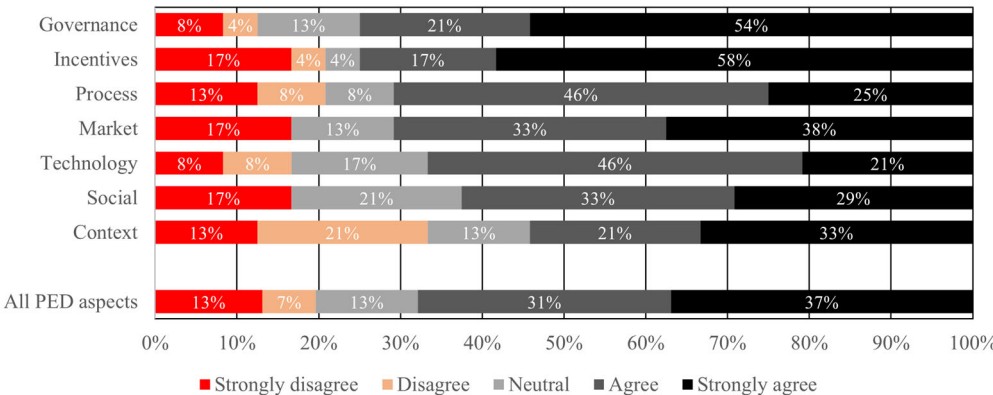

**Figure 5.** Degree of agreement within each PED factor independently, ranked in order of most positive (i.e., combined 'agree' (grey) and 'strongly agree' (black) percentages).

The graph above reveals that there is a range of responses for each topic. Overall, there was a predominant agreement for the seven PED topic categories, with 68% of votes being positive and only 20% negative. The governance and incentives categories drive this overall finding and scored the most positive at 75%. In contrast, the context category elicited more polarized responses with 54% in agreement and 33% in disagreement. When comparing between factors, the scores (see Figure 2) largely matched the rankings (see Figure 5). Governance scored with the greatest agreement and context with the lowest agreement, coinciding with their overall ranking positions of first and seventh, respectively (see Figure 2). Social factors were an anomaly and showed less agreement than might be expected from the overall ranking in third place. Here, social ranked sixth in terms of degree of agreement. This suggests that more focus, knowledge and dissemination is required in this sector, given its overall importance in this and previous studies.

## 5. Discussion

The previous section detailed the seven topics/challenges to the implementation of PEDs identified by the Delphi method, explaining why each is important and what needs to be done to manage them to support the implementation of PEDs. Our contention is that all seven need to be addressed to deliver successful PEDs. Although there are different ways of defining the key factors, these definitions depend to an extent on the perspective of the originators (whether municipalities, engineers, citizens, etc.). Despite this, the literature suggests that there are common and recurring themes, although not all that have been identified here are always present in the literature.

The reviewed literature, which studied one or some of these topics, has not considered the iterative and evolving interactions between them, and the interrelationships between them are not explicitly defined and argued [2,15,74]. Thus, many studies failed to realize or discuss all the interconnected components of PEDs that can also impact the governance and outcomes of the planning and decision-making process. In addition, the significance of context (such as political, cultural and economic), as well as time and external factors, have been largely overlooked. As a result, the literature has provided a quite static presentation of PED systems and does not show a real complex process of transformation and change. To bridge the existing gaps, Table 1 summarizes the identified seven topics/challenges and the interrelation between them, and thereby discusses what is needed (e.g., what resources, capacity, information, etc.) to manage these challenges to support the implementation of PEDs.

**Table 1.** Summary of challenges to PEDs' implementation.

| Categories | Brief Description | Challenge | What Is Needed to Overcome the Challenge? |
|---|---|---|---|
| Governance | System of rule and collaborative activities backed by shared goals that may or may not derive from legal or formally prescribed responsibilities. | There is no governance model or guidance to conceptualize the social and political mechanisms behind implementation of PEDs, clarifying different actors' roles and responsibilities in different stages of planning and decision-making processes. | A systematic understanding of the social and political mechanisms by which the governance system is being processed and a methodological tool to capture these mechanisms. |
| Incentives | Different social, economic and environmental initiatives should be merged in the political system to support deployment of targeted technologies, environmental gains, livability and inclusiveness in districts. | Incentives are contextual and depend on energy technology, market and policies. Providing the right incentives is complex and requires dynamic and iterative monitoring and evaluation systems. | A systematic review and understanding of the context, including policies, technologies, market and local preferences. In addition, it requires a collaborative monitoring and evaluation model (taking into account the process). |
| Social | The energy transmission requires end-users' change from passive consumption to active prosumption and engagement. | There is a lack of engagement culture and infrastructure, lack of public trust and knowledge about technology and process and lack of a joint participatory method, model or roadmap. | Participatory methods and protocols, appropriate education and knowledge sharing, a participatory governance model and market design, right incentives and a funding scheme can increase the local support. |
| Process | Effective and sustainable planning requires dynamic and incremental decision-making approaches to support decision makers to effectively respond to different evolving technologies, policies, actors and processes. | Decision making should consider the dynamic combinations of problems and solutions represented by different actors and exogenous factors. Without an appropriate decision-making approach, different actors can muster power or oppose a decision and cause the stagnation and prolongation of a process. | A joint dynamic, iterative and transparent decision-making approach or model is required that continuously interacts with the planning processes for PEDs and considers the contextual (such as political, economic, cultural and social) factors. |
| Market | Business models that consider the whole process of building, operating and maintaining PEDs can help to coordinate the stakeholders and define their roles for value creation. | There is no predefined single business model for the successful development of a PED; thus, different stakeholders can adopt their own referred business model. This can create complexity, incohesion and inefficiency. | A joint and holistic business model is needed to identify customer segments and revenue streams, which can also suggest policy instruments to mobilize investments. |
| Technology | Combination of innovative solutions can help to get annual net zero energy import, net zero $CO_2$ emissions and annual local surplus production of renewable energy. By balancing demand and supply, we can increase the flexibility, efficiency and resilience of the energy systems. | Balancing between supply and demand is contextual and complex. The lack of standard, affordable seasonal and long-duration energy storage systems for heating, cooling and electricity as well as integration of ICT, big data and automation techniques with each other and different infrastructures are challenging. | Common methodology to quantify the positivity of the district through the combination of several performance indicators. This requires an integrative urban infrastructure that incorporates all urban flows as well as advanced measurement and processing technologies. |

| Categories | Brief Description | Challenge | What Is Needed to Overcome the Challenge? |
|---|---|---|---|
| Context | Any generic framework for implementing PEDs should respond to the specific social, political, climate, economic, etc. locale. | It is very difficult to develop a generic and replicable solution that is adaptative to the contextual characteristics. | A systematic understanding of how different contextual factor can affect different topics, challenges and aspects of implementing PEDs. |

Governance has the most connections to the other factors, reinforcing its primacy as a challenge for PEDs. This challenge is created when a wide spectrum of stakeholders, who share interests, information and resources to develop PEDs, create distributed and pluralistic efforts into the design and implementation, and none of them has complete control over the outcomes. Therefore, all the identified topics/challenges highlight a necessity of collaboration between different actors across sectors and levels. In this regard, managing the identified challenges depends on the development of a practical and evidence-based governance model.

Even though the significance of governance is recognized, there is still a little knowledge about which type of structure and functionality is needed for the design and enactment of PEDs. This requires conceptualization and analysis of the underlying governance practices, addressing which actors, institutions, processes and relational mechanisms at different levels of society can influence the development of PEDs. There is a need to understand how in a world of complexity, conflicting values and interests and human and institutional imperfection of certain governance systems can be effective. To develop a systematic understanding of the social and political mechanisms by which the governance system is being assessed, there is a need for methodological tools that can capture these mechanisms.

According to the findings, the **Social** factors of PEDs elicit a more polarized set of responses related to their significance. This suggests that this is an area that, although widely recognized as important, warrants specific attention, development and integration in a PED framework or toolkit. This calls for development of a bottom-up, democratic and inclusive perspective and culture that can serve as a mechanism to facilitate communication across all levels of the government structures to ensure that the development of PEDs is conducted in a way that benefits a broader range of its inhabitants. In the PED context, attention is increasingly paid to technological innovations and implementation of high-tech solutions, while citizen-centric investigations, i.e., how residents perceive these changes and how their lives or experiences are affected, are limited. In this regard, the necessity of taking people's experience into consideration while still integrating innovations for continuous development of place should be a fundamental principle and reasoning behind the creation of PEDs. To ensure a feedback loop and community-centered processes, a development of a participatory method and culture is essential, which should be integrated with the local planning and decision-making approaches.

Through inclusive and collaborative governance, the planning and decision-making **Process** of PEDs can also be shifted towards ongoing, evolving and transformative learning, where insights from a broad range of stakeholders and disciplines can be garnered. In the shadow of real collaborative networks, certain resilience and adaptability to new changes, ambiguity and long-term consequences of planning and decisions can be provided. Often, the design process cannot be reduced to a simple decision criterion and decision maker, but different perspectives and technological alternatives must be included. For the energy supply of a district where many different stakeholders are involved with different objectives, this idea is particularly true. Especially in projects involving many actors, such as private companies, private persons and other institutions with different interests and expectations, the planning process takes a long time. In the planning of districts where the interests of the public, residents, energy utilities companies, property developers and many others meet,

these complex structures are particularly common. All these must meet the expectations and needs of all stakeholders who additionally require a joint, transparent, objective and structured decision-making process that supports the entire process from the preliminary design to the operation of the supply concept.

The coordination of stakeholders, including policy makers and citizens, is also crucial for designing an appropriate **Market**, referring to the access to finance and the deployment of successful business models in the supply chain. The tools and resource for effective governance will support the market organization as well. Moreover, the supply chain organization is one of the mechanisms that governance needs to process, which is also determined by **Contextual** and social parameters. To that end, the market cannot be seen as an autonomous process, but it is interlinked with the social and contextual aspects of the PEDs. The market design is part of the complexity that requires successful governance to be addressed.

Positivity in the energy balance (**Technology**) of a district requires an inclusive approach that considers all urban sectors and the synergies between them. To establish a common methodology, a decision-making process is needed considering technical, economic, environmental, political and contextual factors. Optimized management of this flexible system requires a governance model that encourages citizen participation and market operations adapted to the generation and consumption profiles. The application of right **Incentives** can help to achieve positivity in the energy balance, leading to solutions that would not be economically viable on a building scale. An appropriate process also supports the realization of this balance, ensuring the transformation of data into information necessary to support decisions. When information is generated as a routine part of PEDs' operations, there is greater likelihood that this information will be used directly to make mid-course corrections and modifications in the implementation phase. Regarding the fact that the learning associated with participating in such a process is experiential and innovative, an integrated planning and decision-making approach enriched with collaborative governance can bring a deep sense of meaningfulness to the work.

The definitions and descriptions of the PED factors that are developed for this work reveal the interrelationship and overlaps between them (see Figure 6). This implies that no single parameter can be considered in isolation of the others and that no factor can be left out in order not to skew an assessment of PEDs.

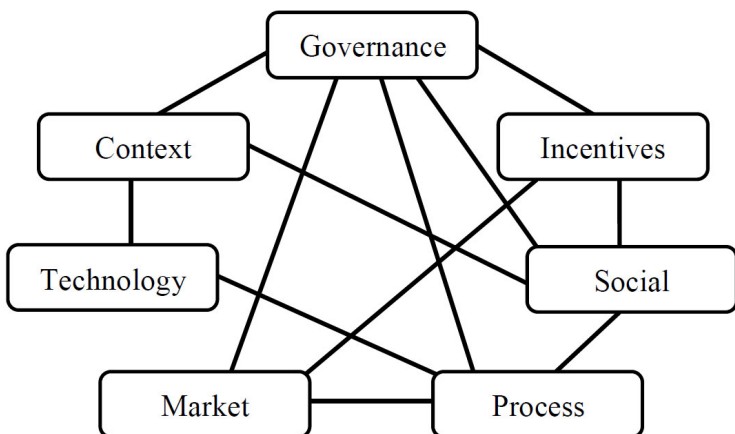

**Figure 6.** Simplified interdependency network of first two interactions for each PED challenge demonstrating the importance for an integrated and holistic PED framework.

## 6. Conclusions

The paper asks what the needs for the implementation of PEDs are. To answer this, we tried to identify the existing challenges through the eyes of different experts who are involved in different projects and initiative related to PEDs. Identifying the challenges would help us to realize the needs (technical and non-technical), as well as

the tools and resources that can address those challenges. The paper used the Delphi method, following by the iterative discussion processes and survey, which allowed for real cooperation, generating feedback mechanisms and meeting the needs for flexibility and adaptability in the planning phases of PEDs. Seven challenges were identified, defined and ranked in this study, named the governance, incentive, social, market, process, technology and context factors. Our finding shows the interrelationship and overlap between the challenges, meaning that managing one challenge to a greater or lesser extent is reliant on the management of the other challenges. In this regard, being confined to only a technical aspect or any individual challenge and failing to consider/scrutinize other effective and interdependent factors/challenges, are two of the main barriers to implementation of PEDs. Therefore, we needed to discover the pattern of their interrelationship and identify which challenge should be dealt with first and why. The outcomes highlighted the significance of the role of governance, which provides the necessary conditions to support all other PED factors. Processes of design and implementation of PEDs are often hierarchical, ranging from international ambitions to local conditions. However, it is evident that top-down diktats are not effective on their own, and that bottom-up, locally relevant narratives are necessary to enhance engagement and success. By developing a holistic, dynamic and iterative collaborative governance model that is context based, it is possible to move forward towards managing the other challenges.

The analysis of the challenges to PEDs implementation and the tools and resources that are needed to address those challenges revealed a common need for systematic understanding of the processes behind those challenges, as well as models and protocols to manage the complexity. Success in developing PED solutions requires a model based on cross-disciplinary collaboration and a co-creation approach that reduces the gap between politics administration, the business sector and society or science and technology, jointly weighing all the challenges that this implementation requires.

None of the reviewed literature and theories is exhaustive in describing each challenge in interconnection with others, and there is no appropriate model, tool or guide that can explain how to manage each challenge considering the influence of other factors. This paper argues that the integration and inclusion of the components of these challenges is necessary to completely understand the real dynamics of PEDs processes. Each factor (political, social, environmental, procedural, economic, technological and contextual) in interaction with other factors can provide a nuanced view of the outcomes of each phase of PEDs processes. Hence, the outcomes are likely to be poorly understood if these factors are looked at exclusively and independently. On the other hand, due to their interaction and interconnection, the boundaries between them are also blurred.

The researchers acknowledge that each of the seven challenges is quite big, and it is difficult to thoroughly explore the significance of each for implementation of PEDs. The contribution of this paper is not to develop the theories/models of these concepts, but to explain their significance in design, planning and implementation of PEDs. Therefore, these issues need to be addressed in greater depth, as arguing what exactly should be done and how it should be done is beyond the scope of this paper.

This paper can be seen as a starting point for exploring the tools and resources that are needed for managing the challenges of PEDs through an integrated and holistic approach of design, planning and implantation. It has been an attempt to explain how the integration of aspects of PEDs can put realism into the study that can overcome the partial explanation that an individual theory, model or tool can provide. In this regard, future research can explore the interdependency between the challenges more deeply and contribute to development of a cross-disciplinary and holistic "conceptual and procedural framework" based on the required tools, guidelines and targets that different districts or cities may need to implement PED. This is the ambition of the PED-EU-NET action.

More research is also recommended to apply such integrated and holistic approaches and to test/integrate different factors to underpin a practical analysis of the complex planning of PEDs and to provide evidence-based results. Even though the collaborators

of this paper do not represent the whole of Europe, this paper targets all the European countries (and beyond), and since the COST Action is still ongoing, the results of this paper can be the basis of the future research works, which can be redeveloped in collaboration with other international programs/initiatives such as Annex 83.

**Author Contributions:** Conceptualization, S.G.K., K.S., T.K., S.S. and N.M.; methodology, S.G.K., K.S., T.K., S.S., M.L., E.G., B.P., T.A., L.M. and N.M.; analysis, S.G.K., K.S., T.K., S.S., M.L., E.G., B.P., T.A., L.M. and N.M.; writing—original draft preparation, S.G.K., K.S., T.K., S.S., M.L., E.G., B.P., T.A., L.M. and N.M.; writing—review and editing, S.G.K., K.S., T.K., S.S., M.L., E.G., B.P. and T.A.; visualization, K.S. and S.S.; supervision, S.G.K., K.S., T.K., S.S. and N.M.; project administration, S.G.K.; All authors have read and agreed to the published version of the manuscript.

**Funding:** This research received no external funding. The Article Processing Charges (APC) was funded by COST (European Cooperation in Science and Technology) under the Action Positive Energy Districts European Network (PED-EU-NET).

**Acknowledgments:** This article is based upon work from COST Action Positive Energy Districts European Network (PED-EU-NET), supported by COST (European Cooperation in Science and Technology, www.cost.eu).

**Conflicts of Interest:** The authors declare no conflict of interest.

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
