# Peer review of "Positive Energy Districts: Identifying Challenges and Interdependencies"

_sustainability, doi:10.3390/su131910551_

Round 1

Reviewer 1 Report

This was an enjoyable read for me. Where the subject is super interesting and the authors have done a good job of capturing the essence of the problem at hand. The new model and how you have presented it is very clear.

There are very few errors in grammar and referencing that I could find: 

18, 207, 227, 240, 253, 312,313, 375, 431, 455, 469, 625

Read threw and check tempus as well as how you have referenced in these parts. 

There were really only three parts that I had any problems with in the text: 

  • 147 where you talk about a responsrate of 15% if I have interpreted the text correctly this actually correlates to 24 people? This in it self is not a great problem, but of course it is not a great number of responses.
  • 405-408 where you state that integrating electricity and thermal energy reduces the flexibility, efficiency and resilience of the system. Here I would actually argue the opposite i.e. having a multitude of prosumers strengthens the system. Here I would argue that you need to strengthen this claim in order for it to be believable. 
  • The only part of your text that I did not particularly like was the conclusion starting at 625. This is more of a summary of what the reader has just gone through. The text is a lot "blander" or less refined and well thought through if you will, than the rest of the text. Here you need to redo and not summarize as much as have an actual conclusion and discussion in and around those conclusions. 

Author Response

Dear Reviewer,

Co-authors and I very much appreciated the positive comments on this manuscript. We have taken them fully into account in revision.

Our responses to your comments are as follows:

Reviewer #1:

  • Point 1: There are very few errors in grammar and referencing that I could find: 18, 207, 227, 240, 253, 312,313, 375, 431, 455, 469, 625 ; Read threw and check tempus as well as how you have referenced in these parts.

  • Response- Thank you for pointing these out. We have revised these lines.

  • Point 2.1: 147 where you talk about a response rate of 15% if I have interpreted the text correctly this actually correlates to 24 people? This in it self is not a great problem, but of course it is not a great number of responses.

  • Response- We agree that it is not a great number of responses, but survey was our secondary and supplementary method, examining the result of the Delphi method. This can also be managed in future studies. We tried to clarify this in different parts of our paper (please see abstract; lines 133-145)

  • Point 2.2: 405-408 where you state that integrating electricity and thermal energy reduces the flexibility, efficiency and resilience of the system. Here I would actually argue the opposite i.e. having a multitude of prosumers strengthens the system. Here I would argue that you need to strengthen this claim in order for it to be believable.

  • Response- We agree that the integration of electrical and thermal energy increases the flexibility and resilience of the system. In this paragraph we wanted to indicate the necessity to integrate these flows in a balanced and optimized way, using advanced control and management techniques to avoid problems that could lead to reductions in the flexibility and efficiency of the system. To make this concept clearer, this paragraph has been modified.

Old paragraph: ‘Integrating both factors together can result in demand and supply balancing issues, which reduces the flexibility efficiency and resilience of the energy systems.’

New paragraph: starting from line 453: ‘Integrating both factors together can result in demand and supply balancing issues, requiring an optimized adjustment through advanced management and control techniques to avoid losses in the flexibility efficiency and resilience of the energy systems.’

  • Point 2.3: The only part of your text that I did not particularly like was the conclusion starting at 625. This is more of a summary of what the reader has just gone through. The text is a lot "blander" or less refined and well thought through if you will, than the rest of the text. Here you need to redo and not summarize as much as have an actual conclusion and discussion in and around those conclusions.

  • Response- We have tried to clarify our main message and contribution in the conclusion part.

Best regards,

Reviewer 2 Report

Dear authors,

thank you fo submitting your paper, I enjoyed reading it a lot. the research design is solid and it was interesting to see the impact of covid period in using the online methods.

some comments from my side:

  1. did you consider the limits from geographical covergare point of view in your panel of experts? according to the affiliations of authors, for instance perspectives from Central Europe are absent.
  2. what are the limitations of your research? please specify in your methodology.
  3. what are the next steps? in the very last paragraph (l. 659-662) you say you are planning to develop a whole framework and other tools. can you be more specific? for instance, h2020 inspiration was aiming at creating a whole research agenda for soil and water research on the level of the EU. this would help the professional and scientific audience to consider your paper in broader perspective and understand its ambitions and how realistic it is.

Author Response

Dear Reviewer,

Co-authors and I very much appreciated the positive comments on this manuscript. We have taken them fully into account in revision.

Our responses to your comments are as follows:

Reviewer #2:

  • Point 1: did you consider the limits from geographical covergare point of view in your panel of experts? according to the affiliations of authors, for instance perspectives from Central Europe are absent.

  • Response: Actually, one of the main incentives to write this paper within the COST Action network was its openness to all countries (geographical coverage), all types of institutions (academia, public institutions, SME/ industry, NGO, European/international organisations, etc.) and all fields of science and technology, which could add extra value. In order to overcome the aforementioned limitation, we used survey and internal meetings at COST to collect the perspectives of other countries/disciplines and sectors, including the Central Europe. To increase the representativeness of more regions of Europe, this expert pool has included perspectives from the representatives of Hungry, Germany, Switzerland and Portugal (see lines 125-145).

Even though the collaborators of this paper do not represent the whole Europe, this paper targets all the European countries (and beyond) and since the COST Action is still ongoing, the results of this paper can be the basis of the future research works and be redeveloped in collaboration with other international programs/initiatives such as Annex 83 (see line 767).

  • Point 2: what are the limitations of your research? please specify in your methodology.

  • Response: In order to highlight the limitations of this research, a new paragraph has been included at the end of section 2.1.

line 146: While initiatives for creating PEDs are launched in many European cities, due to the novelty of PEDs the knowledge in this field is still limited, which has been a methodological challenge in this paper. In addition, the complexity and the multifaceted nature of PEDs make the question of how to implement PEDs unanswered.

  • Point 3: what are the next steps? in the very last paragraph (l. 659-662) you say you are planning to develop a whole framework and other tools. can you be more specific? for instance, h2020 inspiration was aiming at creating a whole research agenda for soil and water research on the level of the EU. this would help the professional and scientific audience to consider your paper in broader perspective and understand its ambitions and how realistic it is.

  • Response: This article analyzes the existing needs when implementing a PED, identifying seven topics or interrelated challenges which require an interdisciplinary interpretation. The methodology used and described in this paper defines and ranks these seven challenges. Therefore, the next steps to follow are based on this information to delve into the type of needs and gaps (technical and non-technical) that must be covered to meet these challenges, identify what type of tools and resources are necessary and define a general framework that contributes to the optimal implementation of a PED.
  • We have tried to clarify our main message and contribution in the conclusion part (please see the conclusion part).

Best regards,

Reviewer 3 Report

The subject is interesting, nevertheless some revisions must be made, namely:

  • The title can be improved: it is too long.
  • The abstract should be more complete, namely with a brief description of the concept of PED. It also may give some details on the methodology , namely regarding the survey, the number of responses,  the nationality of respondents, and where are the PED to be installed.
  • The text should present more bibliographic references.
  • The research paper should present a more deepen literature review on the subject.

Author Response

Dear Reviewer,

Co-authors and I very much appreciated the positive comments on this manuscript. We have taken them fully into account in revision.

Our responses to your comments are as follows:

Reviewer #3:

  • point 1: The title can be improved: it is too long.
  • Response- The title is shortened.

  • Point 2: The abstract should be more complete, namely with a brief description of the concept of PED. It also may give some details on the methodology , namely regarding the survey, the number of responses, the nationality of respondents, and where are the PED to be installed.
  • Response- The abstract is revised; the definition of PED is added, and some words are added. Since survey is our secondary and supplementary method to the Delphi method, we believe it is not necessary to provide the detailed information such as the number of responses and the nationality of respondents. The location of the PED to be installed is not applicable in our research. In addition, since this paper is not quantitative, but rather a qualitative and narrative one, the number of responses does not challenge the validity of the research.

  • Point 3: The text should present more bibliographic references.
  • Response- More references have been added to contextualize the concept of PED, and to support the methodology and the results obtained.

Best regards,

Round 2

Reviewer 2 Report

Dear authors,

many thanks for your replies and taking into account the comments of mine and of the reviewers. I believe the research paper is now ready for publication.